# Early Genomic, Epidemiological, and Clinical Description of the SARS-CoV-2 Omicron Variant in Mexico City

**DOI:** 10.3390/v14030545

**Published:** 2022-03-06

**Authors:** Alberto Cedro-Tanda, Laura Gómez-Romero, Guillermo de Anda-Jauregui, Dora Garnica-López, Yair Alfaro-Mora, Sonia Sánchez-Xochipa, Eulices F. García-García, Alfredo Mendoza-Vargas, Emmanuel J. Frías-Jiménez, Bernardo Moreno, Abraham Campos-Romero, José L. Moreno-Camacho, Jonathan Alcantar-Fernández, Jesús Ortíz-Ramírez, Mariana Benitez-González, Roxana Trejo-González, Daniel Aguirre-Chavarría, Marcela E. Núñez-Martínez, Laura Uribe-Figueroa, Ofelia Angulo, Rosaura Ruiz, Alfredo Hidalgo-Miranda, Luis A. Herrera

**Affiliations:** 1Instituto Nacional de Medicina Genómica, Periferico Sur 4809, Arenal Tepepan, Mexico City 14610, Mexico; acedro@inmegen.gob.mx (A.C.-T.); lgomez@inmegen.gob.mx (L.G.-R.); gdeanda@inmegen.edu.mx (G.d.A.-J.); dgarnical@inmegen.gob.mx (D.G.-L.); yalfaro@inmegen.edu.mx (Y.A.-M.); sonnysanxchez@gmail.com (S.S.-X.); frederikgarcia01@gmail.com (E.F.G.-G.); amendoza@inmegen.gob.mx (A.M.-V.); jfrias@inmegen.gob.mx (E.J.F.-J.); bernsmq@gmail.com (B.M.); 2Researchers for Mexico (Previously Cátedras CONACYT para Jóvenes Investigadores), Av. de los Insurgentes Sur 1582, Crédito Constructor, Benito Juárez, Mexico City 03940, Mexico; 3Centro de Ciencias de la Complejidad, Universidad Nacional Autónoma de México (UNAM), Circuito Centro Cultural S/N, Cd. Universitaria, Delegación Coyoacán, Mexico City 04510, Mexico; 4Innovation and Research Department, Salud Digna, Culiacan 80000, Mexico; abraham.campos@salud-digna.org (A.C.-R.); jonathan.alcantar@salud-digna.org (J.A.-F.); 5Clinical Laboratory Division, Salud Digna, Culiacan 80000, Mexico; jose.moreno@salud-digna.org; 6Hospital General Ajusco Medio, Secretaría de Salud de la Ciudad de México (SEDESA), Encinos 41, Miguel Hidalgo 4ta Secc, Tlalpan, Mexico City 14250, Mexico; jesusortizcirujano@hotmail.com (J.O.-R.); mariana.bg.1994@hotmail.com (M.B.-G.); 7Centro Médico ABC, Av. Carlos Fernández Graef 154, Santa Fe, Contadero, Cuajimalpa de Morelos, Mexico City 05330, Mexico; rtrejog@abchospital.com (R.T.-G.); daguirrec@abchospital.com (D.A.-C.); me80nm@hotmail.com (M.E.N.-M.); 8Laboratorio Arion Genética, Margaritas 440-Bis, Hacienda de Guadalupe Chimalistac, Chimalistac, Álvaro Obregón, Mexico City 01050, Mexico; laura@ariongenetica.com; 9Secretaría de Educación, Ciencia, Tecnología e Innovación de la Ciudad de México (SECTEI), Av Chapultepec 49, Colonia Centro, Cuauhtémoc, Mexico City 06010, Mexico; ofelia.angulo@sectei.cdmx.gob.mx (O.A.); rosaura.ruiz@sectei.cdmx.gob.mx (R.R.); 10Unidad de Investigación Biomédica en Cáncer, Instituto Nacional de Cancerología-Instituto de Investigaciones Biomédicas, Universidad Nacional Autónoma de México (UNAM), Av. San Fernando 22, Belisario Domínguez Secc 16, Tlalpan, Mexico City 14080, Mexico

**Keywords:** SARS-CoV-2, Omicron variant, phylogenetic analysis, haplotype analysis, R346K, odynophagia, dysgeusia

## Abstract

Omicron is the most mutated SARS-CoV-2 variant—a factor that can affect transmissibility, disease severity, and immune evasiveness. Its genomic surveillance is important in cities with millions of inhabitants and an economic center, such as Mexico City. Results. From 16 November to 31 December 2021, we observed an increase of 88% in Omicron prevalence in Mexico City. We explored the R346K substitution, prevalent in 42% of Omicron variants, known to be associated with immune escape by monoclonal antibodies. In a phylogenetic analysis, we found several independent exchanges between Mexico and the world, and there was an event followed by local transmission that gave rise to most of the Omicron diversity in Mexico City. A haplotype analysis revealed that there was no association between haplotype and vaccination status. Among the 66% of patients who have been vaccinated, no reported comorbidities were associated with Omicron; the presence of odynophagia and the absence of dysgeusia were significant predictor symptoms for Omicron, and the RT-qPCR Ct values were lower for Omicron. Conclusions. Genomic surveillance is key to detecting the emergence and spread of SARS-CoV-2 variants in a timely manner, even weeks before the onset of an infection wave, and can inform public health decisions and detect the spread of any mutation that may affect therapeutic efficacy.

## 1. Introduction

Despite the great efforts made by the worldwide scientific and health community to contain the COVID-19 pandemic, a major challenge has been the emergence of variants of concern (VOCs) of SARS-CoV-2. These variants exhibit new features, such as increased transmissibility and immune escape [1], that make epidemic control more difficult [2]. The latest of these variants of concern is B.1.1.529, also known as the Omicron variant. The first case of the Omicron variant of SARS-CoV-2 was identified in South Africa on 2 November 2021, and the variant had spread to 133 countries by 7 February 2022 [3].

The Omicron variant has rapidly become the dominant variant worldwide. Omicron has the highest number of mutations among VOCs; these mutations have been associated with increased transmissibility, resistance to therapy, and partial escape of immunity induced by infections or vaccines [4]. Mexico City has been heavily affected by the COVID-19 pandemic. Two intense epidemic waves occurred in spring 2020 and winter 2020/2021, during which the heaviest toll of hospitalizations and deaths was observed [5,6]. A vaccination campaign began, starting with frontline health care workers and continuing for the rest of the population in descending age order, and was completed in the fall of 2021. In summer 2021, a third wave, associated with the Delta variant, occurred, with reduced case hospitalization and fatality rates. After a period of downward epidemic trends, the first case associated with the Omicron variant was detected on 16 November 2021, in Mexico City, initiating an ongoing fourth wave. As Mexico City is the capital and the largest city in the country, the behavior of the pandemic there has a large impact on national epidemic patterns. Mexico City is the economic and transportation hub of the country [7] and cases can be easily imported and exported to other regions throughout the country. Furthermore, with one of the most important international airports in the country, Mexico City can be a major entry point for new variants from abroad [8].

For these reasons, the control of SARS-CoV-2 spread in Mexico City is paramount to protecting both the local population and people throughout the country. To this end, genomic surveillance is a major tool for the identification and assessment of novel threats [9]. Early detection of the importation and spread of new VOCs, complementing the monitoring of epidemic and public health metrics, may allow policy makers to adapt and rapidly implement strategies to mitigate the epidemic.

In this work, we present an analysis of the early effects of the Omicron variant in Mexico City during its first month of spread. We identify the likely importation pattern, with several importation events followed by local transmission. We show that the growth of the epidemic curve is associated with an increased prevalence of the Omicron variant, at the expense of the previously dominant Delta variant. We also show the high prevalence of the R346K mutation as well as the emergence of different haplotypes during this period. We study differences in clinical presentation to the coexisting Delta variant, finding little evidence of either lessened or increased severity. We discuss the implications of these insights for policy making, highlighting the importance of genomic surveillance associated with epidemiological monitoring in the context of the current pandemic.

## 2. Materials and Methods

Participants. Nasopharyngeal swabs (NPS) were collected from patients for SARS-CoV-2 detection. The study was approved by the ethics and research committee of the Instituto Nacional de Medicina Genómica (CEI/1479/20 and CEI 2020/21); all procedures were in accordance with the ethical standards of the institutional research committee.

Sample collection. NPSs were collected by a trained clinician with a flexible nylon swab that was inserted into the patient’s nostrils to reach the posterior nasopharynx. The swab was left in place for several seconds and slowly removed while rotating. The swab was then placed in 2 mL of sterile viral transport medium. Swabs from both nostrils were deposited in a single viral transport tube, taken to a clinical laboratory, and processed immediately.

SARS-CoV-2 RNA extraction. Total nucleic acid was extracted from 300 µL of viral transport medium from the NPSs or 300 µL of whole saliva using the MagMAX Viral/Pathogen Nucleic Acid Isolation Kit (Thermo Fisher Scientific, Waltham, MA, USA) and eluted into 50 µL of elution buffer.

RT-qPCR detection. For SARS-CoV-2 RNA detection, 5 µL of RNA template was tested using TaqPath master mix (Thermo Fisher Scientific, Waltham, MA, USA). All tests were run on a Thermo Fisher ABI QuantStudio 5 real-time thermal cycler (Thermo Fisher Scientific, Waltham, MA, USA). Samples were selected for inclusion in this study based on viral Ct < 30.

Illumina Sequencing. The libraries were prepared using the Illumina COVID-seq protocol, following the manufacturer’s instructions [10]. First-strand synthesis was carried out with RNA samples. The synthesized cDNA was amplified using ARTIC primers V3 for multiplex PCR, generating 98 amplicons across the SARS-CoV-2 genome. The PCR-amplified product was tagmented and adapted using IDT for the Illumina Nextera UD Indices Set A, B, C, D (384 indices) (Illumina, San Diego, CA, USA). Dual-indexed pair-end sequencing with a 36 bp read length was carried out on the NextSeq 2000 platform (Illumina, San Diego, CA, USA).

Illumina Raw Data Processing and Sequencing Data Quality Assessment. The raw data were processed using DRAGEN Lineage v3.3.4/.5/.6 with standard parameters (Illumina, San Diego, CA, USA). Further samples with SARS-CoV-2 and at least 90 targets detected were processed for lineage designation.

Genetic background description. SARS-CoV-2 Omicron genomes from Mexico City that met the following criteria were downloaded from GISAID [10]: data of sample collection from 14 November to 31 December 2021; only complete sequences with a genome coverage > 95% (n = 783); submission up to 25 January 2021. The Nextclade tool was used to determine amino acid substitutions in the viral genome and generate histogram plots of mutation frequency and phylogenies highlighting mutation occurrence [11].

Variant calling and haplotype analysis. Sequences of Omicron SARS-CoV-2 genomes from Mexico City were downloaded from GISAID. Only complete sequences with an N fraction lower than 0.05 were considered (N = 136). The SARS-CoV-2 reference genome NC_045512.2 was downloaded from NCBI. SNVs per SARS-CoV-2 sequence were obtained with NUCmer, a Perl script pipeline for the alignment of multiple closely related nucleotide sequences [12]. NUCmer was executed with the following parameters: map each position of each query to its best hit in the reference, map each position of each reference to its best hit in the query, and exclude alignments with ambiguous mapping. Variable positions in any SARS-CoV-2 sequence were obtained. Only variable positions observed in at least 13 genomes were further considered. Each SARS-CoV-2 sequence was translated into a compressed representation including only the genotype of the variable positions. A unique combination of alleles, e.g., a unique compressed representation, was considered a haplotype. Haplotypes were used to infer a haplotype network using the haploNet function from the Population and Evolutionary Genetics Analysis System package (pegas) [13]. Briefly, genetic distances (Hamming distance) between all pairwise combinations of haplotypes were calculated using the dist.dna function of the Analyses of Phylogenetics and Evolution package (ape) [3] from this distance matrix, and the minimum spanning tree and the median-joining network were computed using pegas [13]. The number of sequences per haplotype was obtained. The most ancient sequence per haplotype was defined as the one with the earliest date of onset of symptoms.

Phylogenetic analysis. We downloaded the full multiple sequence alignment (MSA) from GISAID on 11 January 2022. This MSA was performed based on 6,900,922 submissions to EpiCoV. The full methods are described on the GISAID website. Briefly, both duplicate and low-quality sequences (>5% Ns) were removed using only complete sequences (length > 29,000 bp). Each sequence was individually aligned to the reference hCoV-19/Wuhan/WIV04/2019, discarding dubious alignments. Sequences that resulted in unique insertions in the reference sequence and that occurred more than once were used as an initial set of sequences for multiple sequence alignment, reducing each contiguous stretch of NNNs into a single letter N. The remaining sequences were aligned to the resulting alignment. From this MSA, we calculated the Hamming distance from any sequence of our collection to any of the MSA sequences. We selected the closest sequences (Hamming distance ≤ 10) and calculated the best-scoring maximum likelihood tree using the model GTR + Optimization of substitution rates + Optimization of site-specific evolutionary rates (50 bootstrap analysis) via raxmlHPC [14]. iTOL was used for tree visualization [15].

Clinical data acquisition. We used Sistema Nacional de Vigilancia Epidemiológica (SINAVE), a COVID-19 database hosted by the federal health authorities, to explore demographic and clinical information for 350 patients. Briefly, this database contains clinical information captured during an initial interview, including age, sex, 20 symptoms, 10 comorbidities, date of symptom onset, hospitalization status, death date (if applicable), and vaccination date. Since the Mexican vaccination strategy considers the use of several different vaccines, for the purposes of this work, we consider a patient to be fully vaccinated 14 days after the last dose specified for the vaccine used (monodose or two-dose regimes).

Sequenced data were matched to clinical information using an internal lab ID. All cases from samples with symptom onset from epidemiological weeks 2021-48 to 2021-52 identified in medical units in Mexico City were kept. Samples not belonging to the Delta and Omicron VOCs were discarded.

Comparisons between Delta and Omicron populations. We performed statistical analyses to assess whether the Delta and Omicron populations exhibited differences in any variable of interest. We used a combination of statistical tests based on the comparisons of interest:

For numerical variables (age, number of comorbidities, number of symptoms), we used a two-sided *t*-test.

For categorical variables, we used Fisher’s exact test to test for overrepresentation of the feature (equivalent to difference in proportions) within any of the variants. For multivariate analyses, we performed logistic regressions of the form:$$p\left(\mathrm{VOC}\right)=f\left(features_{i}\right)$$

The equation represents the logistic model to represent that the variant of concern (VOC) label (Delta or Omicron) is a function of the set of features in the model (either comorbidities or symptoms). In addition, we obtained from SINAVE all registered SARS-CoV-2-positive cases with and without variant sequencing data (n = 4926).

All statistical analyses were performed using R version 4.1.2 (1 November 2021)—”Bird Hippie”. Data manipulation and visualization were performed using Tidyverse tools [9], and statistical tests were performed using the Tidymodels suite.

## 3. Results and Discussion

### 3.1. Phylogenetic Analysis

We performed a phylogenetic analysis to infer the possible origin of Omicron SARS-CoV-2 in Mexico. We analyzed all Omicron SARS-CoV-2 samples collected in Mexico together with their closest worldwide relatives. The best-scoring maximum likelihood (ML) phylogeny is presented in Figure 1. We can observe that several exchange events between Mexico and the rest of the world have occurred, since sequences collected in Mexico are located all over the phylogeny. We can also observe that several independent exchange events have occurred between Mexico and the USA (black arrows). However, the direction of these exchange events cannot be interpreted from this analysis. Additionally, several exchange events cannot be associated with a specific country. For example, Figure 1B shows a monophyletic group formed by sequences from Mexico, the United States of America (USA), the United Kingdom (UK), France, Germany, and Ecuador, and Figure 1C (lower clade) shows a monophyletic group formed by sequences from Germany, Israel, the USA, Mexico, France, and the UK. Interestingly, most exchange events between Mexico and the rest of the world are associated with groups of sequences that present a low distance to their foreign counterparts; however, there is one group of sequences collected in Mexico that have largely diverged from the rest of the world (gray arrow), displaying shorter distances between them compared to any other distances. The origin of this monophyletic group of sequences is unclear. Interestingly, most of the exchange events have associated sequences from Mexico City and from the rest of the country (Figure 1D); however, the most divergent monophyletic group contains sequences from only Mexico City (gray arrow). The observed pattern suggests multiple independent exchange events between Mexico and the rest of the world in the first month of Omicron occurrence in Mexico. Concurrently, it also suggests a single event followed by local transmission that gave rise to most of the diversity of Omicron SARS-CoV-2 sequences observed in Mexico City. The scenario depicted in Mexico is not rare, since independent importations of specific SARS-CoV-2 variants have been described in the literature [11,12], as have superspreading events following recurrent importations [13]. This behavior can be associated with frequent human movement between and within country borders.

### 3.2. Identification of the Omicron Variant

The first case of the Omicron variant in Mexico was detected on 16 November 2021, in Mexico City, and, as of 31 December 2021, 783 cases of Omicron were detected by next-generation sequencing in Mexico City. According to the pangolin lineage, 57% were BA.1 and 43% BA.1.1 [3].

At week 46, the Omicron variant represented 2% of cases (BA.1 50% and BA.1.1 50%), while the Delta variant represented 98%. Then, in week 50, the Omicron variant increased rapidly, reaching 65% (BA.1 66% and BA.1.1 34%), while Delta dropped to 35%. At week 52, the Omicron variant accounted for 88% (BA.1 51% and BA.1.1 49%) and the Delta variant accounted for 12% (Figure 2A). The increase in positive cases was possibly due to the increased prevalence of the Omicron variant; Figure 2B shows that SARS-CoV-2-positive cases increased in the periods where there was a higher prevalence of Omicron samples detected by sequencing. At the same time (week 52), the Omicron variant had different prevalences throughout the Americas, e.g., 59.29% in the USA, 37.57% in Brazil, 60.37% in Colombia, and 45.8% in Argentina, while the prevalence was 69.33% in the UK and 98.87% in South Africa [3].

Never before had a VOC displaced other variants so quickly in Mexico City. In 2021, variant B.1.1.519 was totally displaced by Alpha, Gamma, and Delta in four months [6], while the Omicron variant displaced Delta in only one month. There was a correlation between the growth of the epidemic curve and the proportion of Omicron variants (Figure 2C). The increase in SARS-CoV-2-positive cases began on 29 December 2021, in Mexico City, a month and a half after the first detection of Omicron by sequencing, which speaks to the importance of SARS-CoV-2 genomic surveillance programs for public health decision making [14].

### 3.3. Omicron Genetic Background

Of the 783 Omicron genomes sequenced from 16 November 2021 to 31 December 2021 in Mexico City, the region coding for the spike protein contains 50 nonsynonymous mutations, 19 of which are located in the receptor-binding domain (RBD). Figure 3 shows the most frequent amino acid substitutions in the RBD (G339D, R346K, K417N, S371L, S373P, S375F, N440K, Q498R, and N501Y). The frequency of all substitutions, deletions, and insertions is shown in detail in Appendix A, respectively. N501Y is present in the Alpha, Beta, and Gamma variants, and this substitution increases binding affinity to angiotensin-converting enzyme receptor 2 (ACE2), playing an essential role in the higher rate of transmission of SARS-CoV-2 variants [15]. K417N/T is present in the Alpha and Gamma variants, and both mutations facilitate immune escape for monoclonal antibodies (bamlanivimab/LY-CoV555) [16], escape from neutralization by convalescent plasma [17], and escape by sera from BNT162b2-vaccinated individuals [18]. These results indicate that Omicron has a significant immune escape from existing protection established by virus infection or vaccination, most likely due to the accumulation of the mutations described above [19,20].

We performed a hierarchical clustering of the prevalence of the most frequent substitutions in the spike protein in the top three countries that generated the most Omicron genomes per continent. This analysis shows that the prevalence in substitutions is similar between Mexico City and Mexico outside of Mexico City, Germany, and Brazil, since they clustered together.

We found that the substitutions R346K, K417N, and N440K were more prevalent in Mexico City, Mexico outside of Mexico City, the USA, and Japan (42–52%). In addition, a group of five substitutions (A67V, del69/70, T95I, G142D, del143/145) had the highest prevalence values in all countries analyzed (92–100%), except in the United Kingdom and India (60–71%).

On the other hand, the prevalence of eleven substitutions (S371L, S373P, S375F, S477N, T478K, E484A, Q493R, G496S, Q498R, N501Y, and Y505H) was lower in Mexico City and Mexico outside of Mexico City, Israel, the United Kingdom, and India (32–70%) than in Australia, Canada, Japan, and the USA (91–100%) (Figure 4).

### 3.4. Prevalence of the R346K Substitution

The prevalence of R346K was 42% in Mexico City and 46% in the rest of the country; other countries with high prevalence were the USA (42%) and Japan (52%). Interestingly, samples with R346K in Mexico City clustered in a monophyletic branch in a phylogenetic analysis (Appendix A). R346K was presented for the first time in the Mu variant B.1.621 [21] and is now present in the Omicron variant. This substitution has clinical importance in the therapy of COVID-19. Neutralization studies on the B.1.1.529+R346K pseudovirus showed that 18 of the 19 monoclonal antibodies (mAbs) tested lost neutralizing activity completely or partially [22]. There are mAB-based therapies that have demonstrated efficacy in neutralizing Omicron+R346K, such as STI-9167 from Sorrento Therapeutics [23], sotrovimab by GSK, and Vir Biotechnology [24].

### 3.5. Haplotype Network

Virus evolution can be represented by a haplotype network. Any node in a haplotype network represents a virus haplotype, i.e., a unique combination of alleles present in at least one patient sample (Appendix A). The weight of an edge represents the number of mutations between the two connected haplotypes, i.e., the number of mutations required to transition from one haplotype to another. The final haplotype network is the maximum parsimony representation of virus evolution, as the total weight of the network is minimized during the construction of a haplotype network.

We built a haplotype network from Omicron SARS-CoV-2 sequences from Mexico City (Figure 5). Some haplotypes generates a large number of descendent haplotypes, forming star-like structures in the network; this phenomenon has been seen in previously reported haplotype networks from SARS-CoV-2 and other viruses [25,26]. The large nodes represent common haplotypes, as they are observed in a large number of patients. The color is proportional to the fraction of patient samples that were fully vaccinated for each haplotype. The white nodes belong to haplotypes with 100% nonvaccinated patients, and dark purple nodes belong to haplotypes with 100% fully vaccinated patients. The haplotypes marked with red arrows correspond to common haplotypes, as they are present in six and four patients, respectively, all of whom were fully vaccinated. Further investigation is required to understand this relationship. There were 64 haplotypes composed of solely fully vaccinated patients, 32 of which (50%) corresponded to internal nodes; on the other hand, there were 21 haplotypes composed of solely unvaccinated patients, and, similarly, 11 of them (0.53%) corresponded to internal nodes. Internal nodes belong to haplotypes that diversified into other virus haplotypes. We observed that most haplotypes were present in only one patient, making any interpretation difficult. Moreover, in previous studies, only marginal differences in haplotype distributions have been observed between countries with low and high vaccination rates [27].

### 3.6. Epidemiology and Clinical Associations

As previously mentioned, we observed an increase in the prevalence of the Omicron variant over the previously predominant Delta variant in the span of ~1 month. During this transition, basic population descriptors (as seen in Figure 6) of the infected population, such as age distribution, sex distribution, number of reported comorbidities, and number of reported symptoms at the time of diagnosis, remained virtually unchanged, with no significant differences between the variant groups (see Table 1 and Appendix A).

In terms of vaccination, 66% of our cohort was composed of vaccinated individuals. It should be noted that in this cohort of sequenced samples, only five patients required hospitalization and only one death was reported on the last day of the analysis period. Due to this reduced number of adverse outcomes, our analyses are not able to test whether there is an association between the Omicron variant and hospitalization or death outcomes in the analysis period, much less to model the effect of vaccination status as other groups have [28]. We should emphasize that the data analyzed in this work, which are essentially early-stage variant introduction data, neither support nor reject the notion of a difference in the likelihood of adverse outcomes associated with the Omicron variant becoming the dominant variant, particularly in the context of policy making and mitigation strategies; the intrinsic severity of Omicron is still an open question that should be addressed with great care [29].

We assessed whether any of the measured comorbidities that are routinely reported in the Mexican COVID-19 case recording protocol were differently represented in the Omicron or Delta populations. The logistic regression model showed that none of these comorbidities was associated with a specific variant. These results suggest that risk assessment based on comorbidities and associated clinical risk predictors currently used [5,30] may remain valid for the Omicron variant.

In terms of symptomatology, a logistic regression model (see Table 2) showed two symptoms as significant with respect to differentiation between the Delta and Omicron variants. These are dysgeusia (distortion of taste) and odynophagia (pain with swallowing). The former is more associated with the Delta variant, while the latter is more associated with the Omicron variant. In Figure 7, we illustrate this association by showing the percentage of all recorded cases in Mexico City (regardless of sequencing status); a drop in dysgeusia is clearly observable as Omicron becomes more dominant. While there have been reports in the media and in the recent medical literature [31] of changes in the symptomatology associated with the Omicron variant, the modest results of our regression model suggest that a diagnosis of COVID-19 caused by the Omicron variant through symptoms alone may be unfeasible, highlighting the need for proper genomic surveillance. However, monitoring signals, such as the drop in reports in dysgeusia, may provide early warning signs of the introduction of the variant to new populations and trigger assessment using appropriate genomic tools. Appendix A shows the detailed clinical data for the 350 samples used in this analysis.

We also analyzed the difference between the Ct distribution for the samples classified as either Delta or Omicron for each of the markers amplified during the qRT-PCR test. The distributions are shown in Figure 8. We observed a sharp difference in the amplification of the spike gene, in accordance with the detection kit manufacturer’s observation of a dropout of the S-gene target associated with the Omicron SARS-CoV-2 variant. This dropout was first identified in the Alpha variant. It is caused by the 69–70 del mutation of the S gene that interferes with the amplification of the S-gene target by the TaqPath Thermo Fisher Scientific PCR Kit. We note that of the Omicron samples, only eight samples had amplification in the Spike with Ct values 16 to 33, of which five samples have a true exponential amplification curve in the spike gene in the PCR, even though they have the 69–70 deletion in Spike detected by sequencing. The remaining three samples did not have a true amplification curve for the spike gene in PCR. Our results suggest that a qRT-PCR test could be an initial and efficient approach to propose the variant classification of a patient sample, at least at this moment in the course of the pandemic, where practically all samples belong to either Delta or Omicron variants. In addition, epidemiological surveillance and monitoring are important to rapidly detect unexpected behaviors during qRT-PCR amplification that could be associated with the emergence of new variants.

We performed a *t*-test on the nucleocapsid and orf1ab Ct distributions to compare the Ct mean between the Delta and Omicron variants. We found a nonsignificant difference (*p*-value = 0.0585) in the case of the nucleocapsid marker and a significant difference in the case of the orf1ab marker (*p*-value = 0.0171). Although viral load and Ct cannot be directly correlated [32,33], the possible clinical significance of the observed change remains to be investigated.

## 4. Conclusions

Mexico City is the most populated city in Mexico as well as the political and economic center of the country. It is also one of the largest tourist and commercial entry points. Epidemiological surveillance in high-movement urban regions, such as Mexico City, is important for the timely detection of the appearance and propagation of new SARS-CoV-2 variants even weeks before the highest number of cases is reported.

In this study, we have characterized the early stage of what has become the fourth wave of COVID-19 in Mexico City, marked by the dominance of the Omicron variant. Phylogenetic analyses show that Omicron likely had several importation events, with one of them giving way to the dominant transmission tree. We have shown that Omicron in Mexico City has genomic variability, giving way to many haplotypes as well as cases exhibiting mutations with clinical significance. After its importation, Omicron displaced the previously dominant Delta variant in the space of one month.

During this first month, in which the Omicron and Delta variants coexisted in Mexico City, we observed few differences in clinical presentation: only differences in symptoms of odynophagia and dysgeusia were statistically significant. However, it should be noted that during this period hospitalizations and deaths were at historical minimums; therefore, we were unable to properly assess whether risk for any of these outcomes was different between the Delta and Omicron variants. This is not, however, evidence of equal clinical presentation, which may be assessed in future retrospective studies after the wave subsides. Therefore, it is important to continue monitoring the pandemic behavior to detect patterns that could inform public health decisions and guide decision makers. As the pandemic is a highly dynamic phenomenon, it is important to continue genomic surveillance to detect the propagation of any mutation that could affect treatment selection or effectiveness as well as changes in the clinical features and public health indicators associated with the spread of the virus.

## Figures and Tables

**Figure 1 viruses-14-00545-f001:**
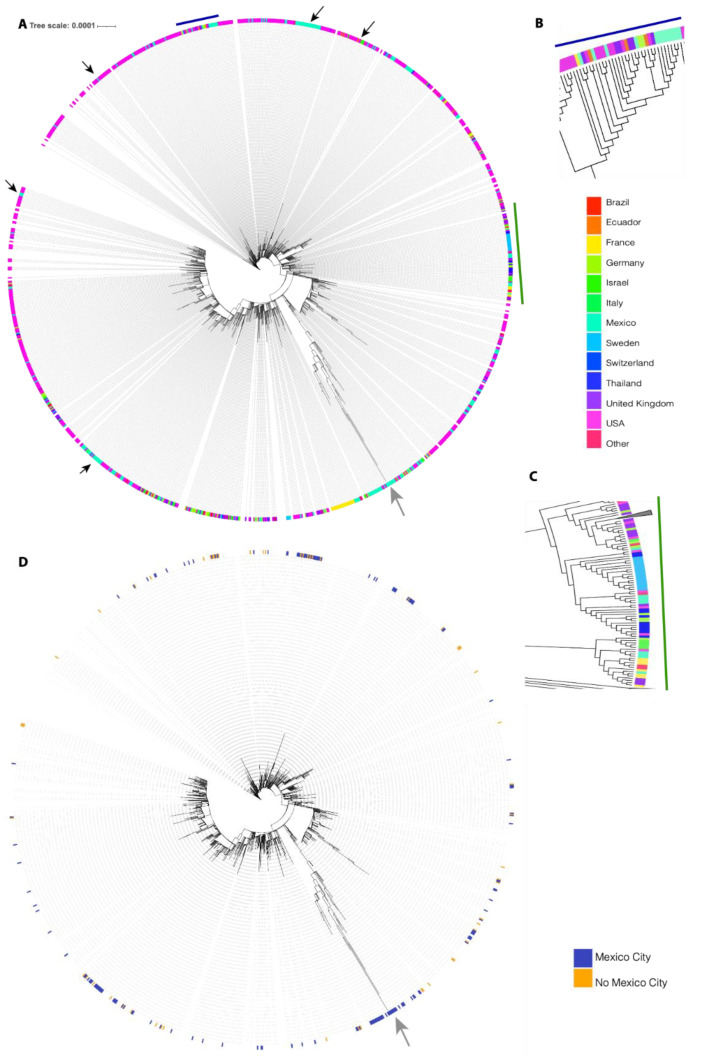
The best-scoring ML phylogeny of Omicron SARS-CoV-2 samples collected in Mexico together with their closest worldwide relatives. (**A**) The best-scoring ML phylogeny of all sequences is presented. Black arrows indicate exchange events between the USA and Mexico. The gray arrow displays a group of sequences collected in Mexico that have largely diverged from the rest of the sequences. (**B**) A zoom on the region indicated by the blue line. (**C**) A zoom on the region indicated by the green line. Colors represent countries in which the analyzed sequences were collected. (**D**) Blue represents sequences collected in Mexico City; orange represents sequences collected in Mexico but not in Mexico City.

**Figure 2 viruses-14-00545-f002:**
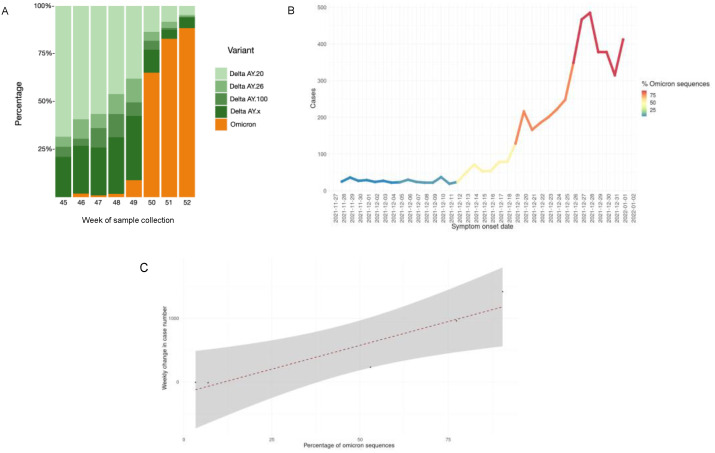
(**A**) Prevalence of the Omicron variant in Mexico City from 16 November to 31 December 2021 (week 45 to week 52). (**B**) Percentage of Omicron variant cases and symptom onset dates. (**C**) Correlation plot between the weekly growth rate of the epidemic curve and the proportion of Omicron variants in 4296 samples from SINAVE.

**Figure 3 viruses-14-00545-f003:**
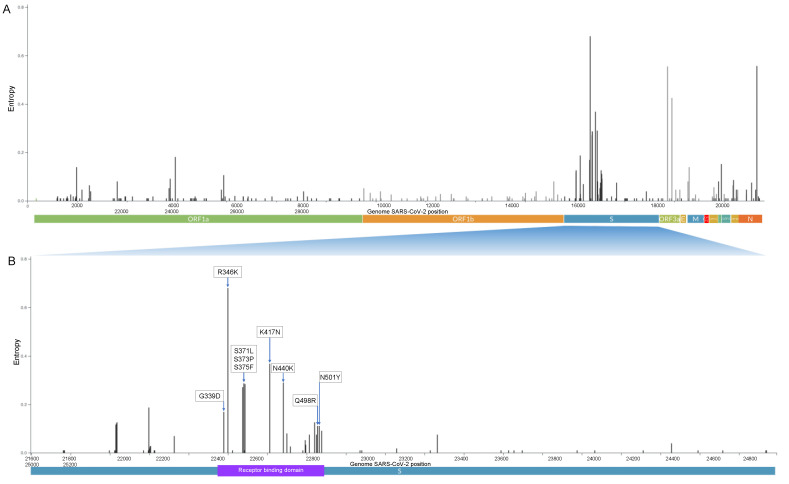
(**A**) Genome map of the SARS-CoV-2 Omicron variant with the most representative amino acid substitutions in 783 Omicron genomes in Mexico City with genome coverage >95%. Whole-genome SARS-CoV-2. (**B**) Spike protein with receptor-binding domain. The y-axis corresponds to the entropy calculated by the Nextclade tool.

**Figure 4 viruses-14-00545-f004:**
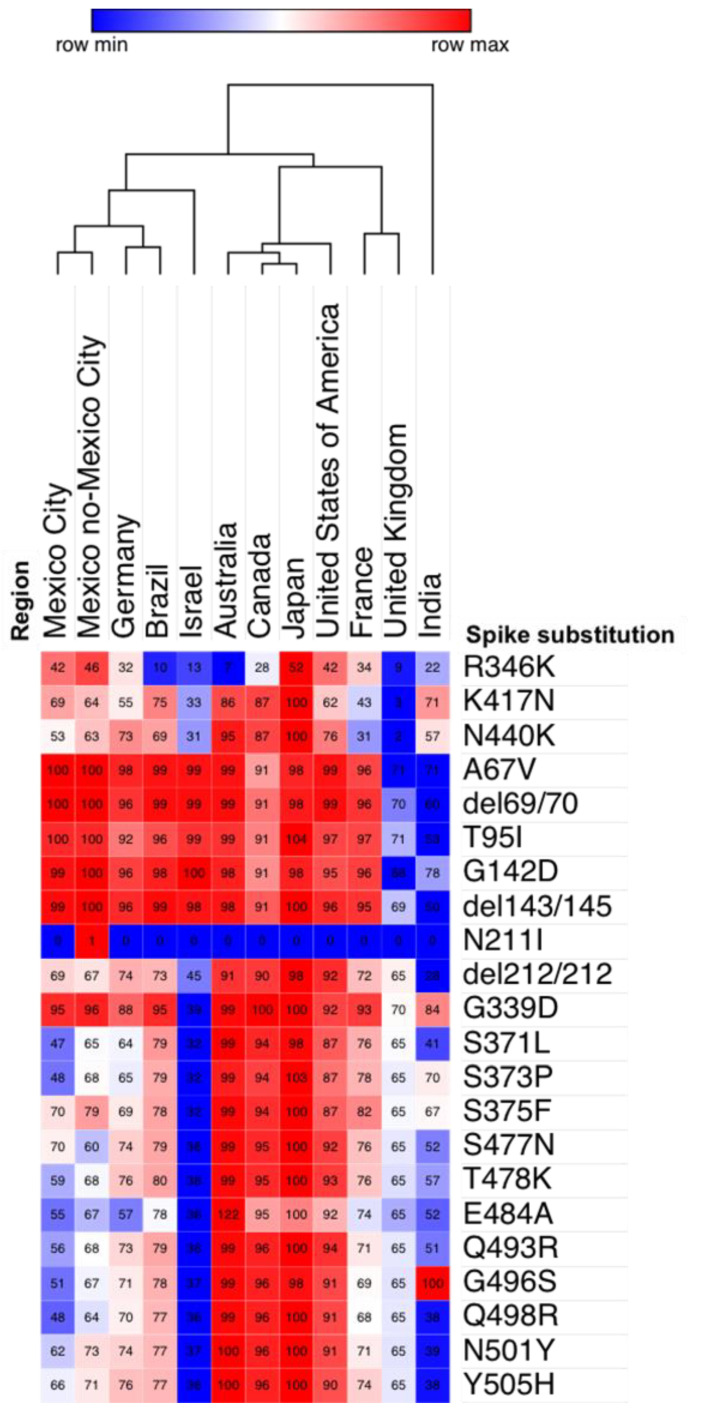
Hierarchical clustering of the prevalence of amino acid substitutions in the spike protein among different countries in samples collected from 16 November to 31 December 2021, obtained from GISAID. We use one minus the Pearson correlation and clustering by region and the average as the linkage method. The scale represents the prevalence by each substitution in countries. Each cell contains the prevalence of the substitution as a percentage.

**Figure 5 viruses-14-00545-f005:**
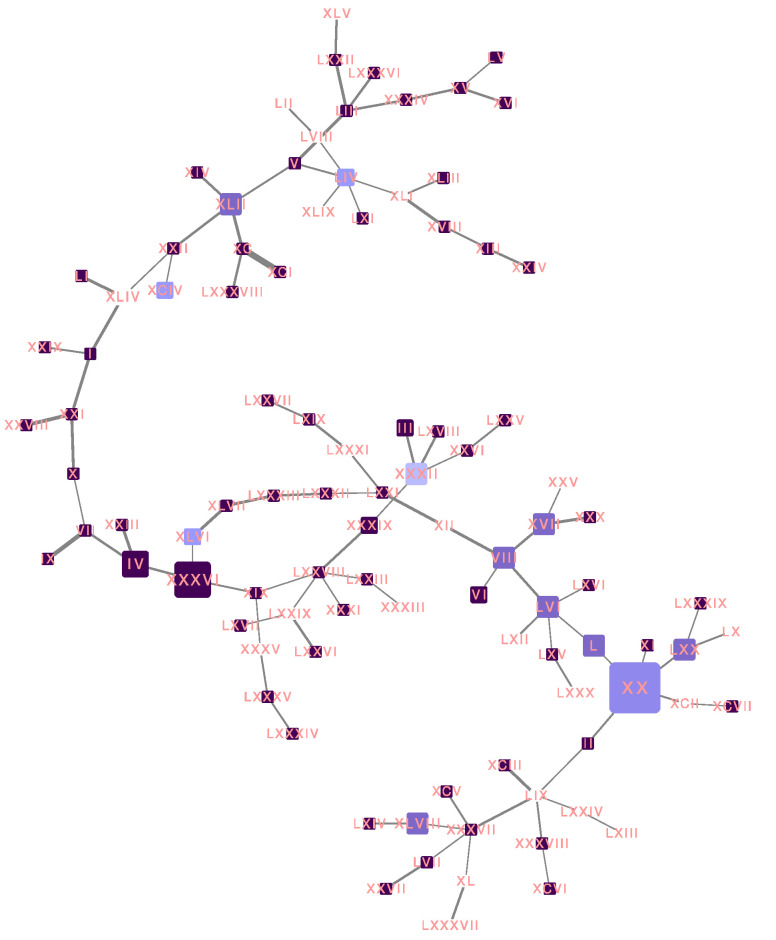
Haplotype network for all Omicron SARS-CoV-2 sequences from Mexico City. The size of each node is proportional to the number of samples that belong to that haplotype. The color represents the fraction of patient samples that were fully vaccinated for each haplotype. The white nodes belong to haplotypes with 100% nonvaccinated patients, and dark purple nodes belong to haplotypes with 100% fully vaccinated patients. The width of the lines is proportional to the number of mutations between two haplotypes.

**Figure 6 viruses-14-00545-f006:**
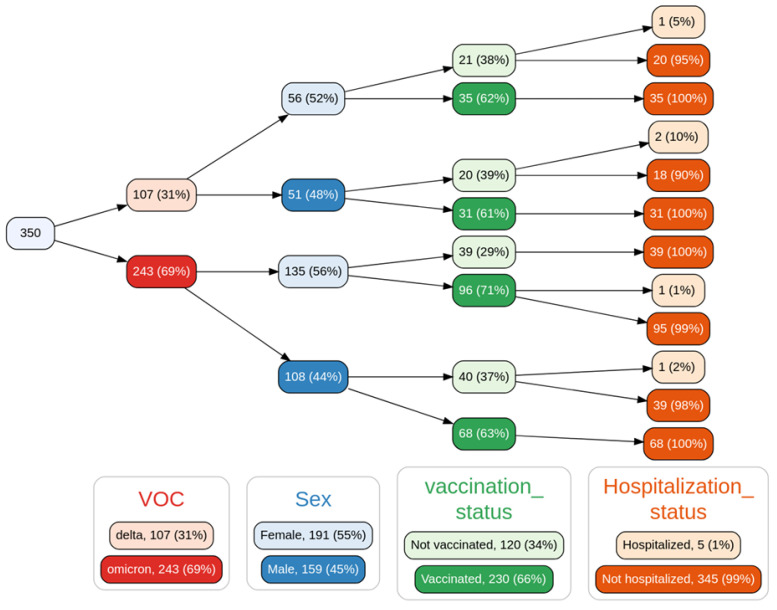
Variable tree with the spread of samples based on variant of concern (VOC), sex, vaccination status, and hospitalization status. Not shown: a single sample in the Delta->male->not vaccinated->hospitalized branch that was the only sequenced case from a patient who died in the analysis period. Only 350 of the 783 samples sequenced were able to provide the clinical data for this analysis.

**Figure 7 viruses-14-00545-f007:**
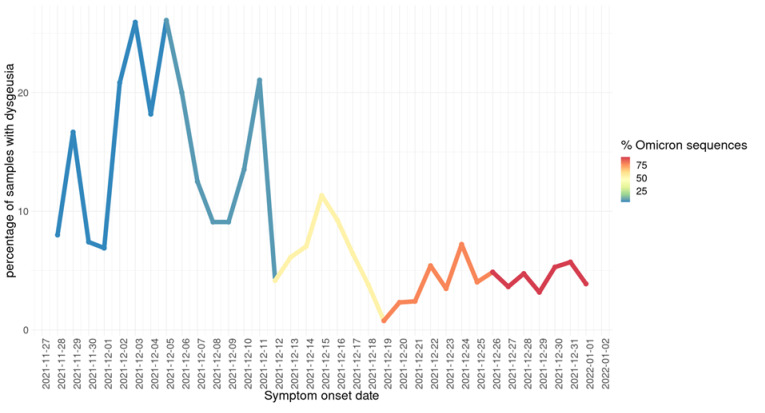
Daily fraction of all COVID-19-positive patients detected in Mexico City who reported dysgeusia as a symptom (by symptom onset date). Line color indicates the (weekly) percentage of sequenced samples identified as Omicron detected by genomic surveillance.

**Figure 8 viruses-14-00545-f008:**
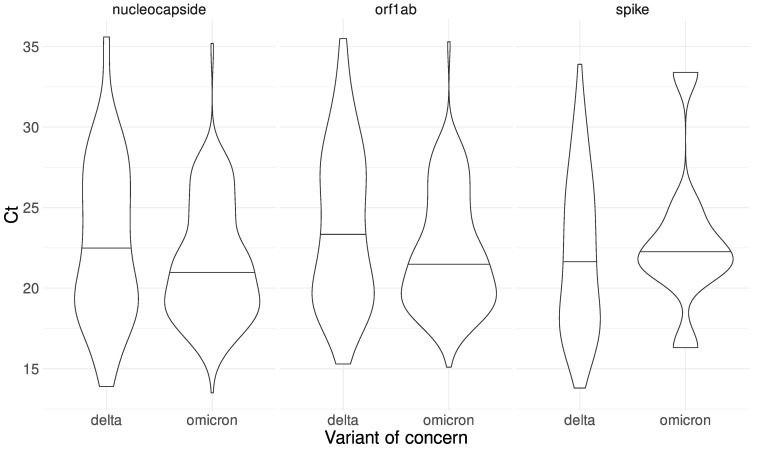
Ct distributions for each gene (nucleocapside, orf1ab, and Spike) used during the RT-qPCR test for Delta and Omicron SARS-CoV-2 samples. A two-sample *t*-test was performed for each marker. The corresponding *p*-value is shown in each case. The dashed lines represent the mean for each distribution.

**Table 1 viruses-14-00545-t001:** *p*-values of the comparison (Fisher’s exact test) between the Delta and Omicron variants.

Variable	*p*-Value
Sex	0.6414
Vax	0.3285
Hospitalization status	0.1692
Age	0.1801
Number of comorbidities	0.7300
Number of symptoms	0.8343

**Table 2 viruses-14-00545-t002:** Logistic regression variants as a function of symptoms (Delta variant as baseline). Predictors with *p* < 0.05 are shown.

Term	Odds Ratio	*p*-Value
Intercept	2.01	0.0021
Odynophagia	2.10	0.0090
Dysgeusia	0.103	0.0009

## Data Availability

All genetic data is available at the GISAID Initiative web page (https://www.gisaid.org/, accessed on 25 January 2022).

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
