# Peer review of "Early Genomic, Epidemiological, and Clinical Description of the SARS-CoV-2 Omicron Variant in Mexico City"

_viruses, 2022, doi:10.3390/v14030545_

Round 1

Reviewer 1 Report

Remarks to the Author:

In the manuscript by Cedro-Tanda et al (viruses-1618479), the authors analyzed the Omicron variants spreading in Mexico City from mid-November to the end of 2021. The data indicated that Omicron became the dominant variant circulating in Mexico City one month after the first case reported in Mexico. The authors did a multiple comparison analysis for Omicron in Mexico City by phylogenetic and haplotype analysis. These analyses provide first description of origin, mutation burden and prevalence of Omicron in Mexico City. However, the data indicated that there are no significant clinical differences between the populations infected by the Delta or Omicron variants. Now we know that Omicron exhibited increased transmissibility and changed clinical disease presentation, which is not the case in this study. It could be the real situation for the early spreading of Omicron in Mexico City or due to the small sample size for the analysis.

Overall, the current version of this manuscript should be significantly improved before it can be considered qualified to be published on Viruses.

There are several experimental and discussion questions that should be addressed as detailed below.

Specific comments:

  1. In Figure 2B, the authors analyzed the percentage of Omicron sequences and symptom onset date. As shown in Figure 2C, more than 75% of sequences are Omicron in the last several days. There are more than 783 omicron cases in Mexico City, not just 783 cases as authors claimed. Clearly description for this figure is needed.
  2. In Figure 5 figure legend didn’t clarify which color is vaccinated or unvaccinated.
  3. In Figure 6, the symptom information should be included. The authors showed 243 of 783 Omicron cases, and only 107 cases for Delta variants, which is not enough to give comprehensive understanding of Omicron clinical presentation.
  4. In Figure 8, the author showed the Ct value of different genes of SARS-CoV-2. The spike gene ct value between delta and omicron is too big to be true. Besides that, which RT-PCR kit(s) and what primer sequences were used for the RT-PCR? It was reported that the Thermol Fisher Scientific RT-PCR kit cannot detect the Omicron Spike gene due to the del69-70 in the spike.
  5. There is no clear conclusion for this study, the authors emphasize the importance of genome surveillance for the variants, and finally claimed that it’s important to continue monitoring the pandemic behavior. The authors should include more data to provide the more accurate conclusion for this study.
  6. There is no discussion for this study, discussion is an important part of a research article.
  7. There are some typo or labeling issues, eg, Figure4 figure legend. “In spike proteinspikes” should be “in spike protein”. There I s no labeling for Figure 8 panel A-C.

Author Response

Response to Reviewer 1 comments

Point 1. In Figure 2B, the authors analyzed the percentage of Omicron sequences and symptom onset date. As shown in Figure 2C, more than 75% of sequences are Omicron in the last several days. There are more than 783 omicron cases in Mexico City, not just 783 cases as authors claimed. Clearly description for this figure is needed.

Response: We thank the reviewer for this comment. The 783 samples used to construct Figure 2A belong to sequenced cases deposited in GISAID, which does not represent all cases from Mexico City, as only a small fraction is sequenced. To infer that the increase in positive cases was due to the increased prevalence of the Omicron variant, Figure 2B was constructed, which shows that SARS-CoV-2 positive cases increased in the periods where there was a higher prevalence of omicron samples detected by sequencing. This was clarified in the manuscript in the 324-327 lines.

Point 2. In Figure 5 figure legend didn’t clarify which color is vaccinated or unvaccinated.

Response: We have detailed in the figure caption the color to which the vaccinated and unvaccinated samples belong, placed in the lines 434-439.

Point 3. In Figure 6, the symptom information should be included. The authors showed 243 of 783 Omicron cases, and only 107 cases for Delta variants, which is not enough to give comprehensive understanding of Omicron clinical presentation.

Response: Of the 783 samples reported as Omicron from GISAID, we only had access to clinical data for 243 Omicron and 107 Delta cases, which are shown in Figure 6, this was clarified in lines 459-460. We have incorporated a supplementary file containing the anonymized data of these cases discussed in the manuscript (lines 493-494).

The goal of this manuscript is to provide an early description of the fourth wave in Mexico City, currently ongoing, which is dominated by the omicron variant. In this regard, the analyzed timeframe comprises the period from the introduction of the variant to it becoming the dominant (as detected by genomic surveillance). With this in mind, we agree that a full description of omicron's epidemiological profile is not possible with just this data; however, we should emphasize that this was not the intent of the work. Rather, we were interested in having the most complete description of this early timeframe, in which both delta and omicron variants coexisted in Mexico City, to search for signals of changes in (population or individual) risk associated to omicron.

Point 4. In Figure 8, the author showed the Ct value of different genes of SARS-CoV-2. The spike gene ct value between delta and omicron is too big to be true. Besides that, which RT-PCR kit(s) and what primer sequences were used for the RT-PCR? It was reported that the Thermol Fisher Scientific RT-PCR kit cannot detect the Omicron Spike gene due to the del69-70 in the spike.

Response. You are correct, the difference in S gene amplification between Delta and Omicron is due to a technical issue as we use the Thermo Fisher Scientific TaqPath PCR kit, which does not have the ability to detect the S gene as the design of their patented primers target the 69-70 spike deletion. This has been detailed in the 507-508 lines.

Point 5. There is no clear conclusion for this study, the authors emphasize the importance of genome surveillance for the variants, and finally claimed that it’s important to continue monitoring the pandemic behavior. The authors should include more data to provide the more accurate conclusion for this study.

Response. We agree with the reviewer in that the conclusion of the manuscript in its current form, does not properly convey the findings of this descriptive work. In this regard, we have modified the conclusion to include the following key findings: 1) Omicron likely had several importation events, with one of this dominating later. 2) Omicron in Mexico City exhibited many haplotypes and some mutations with clinical significance. 3)Omicron displaced delta in the space of 1 month. During this month, we did not (yet) observe increases in hospitalization and deaths, and found no differences in hospitalization and death for the sequenced samples and 4) Clinical differences between delta and omicron variants were few; only odynophagia and dysgeusia differences were statistically significant.

Additionally, we have modified the conclusion to highlight what has been discussed in other sections of this reply: mainly, that this is a descriptive analysis of the early stages of this fourth wave, with the aim of gaining early insights; and that future retrospective work may identify further features of the impact of the omicron wave in Mexico City.  This has been detailed in the 530-565 lines.

Point 6. There is no discussion for this study, discussion is an important part of a research article.

Response: As the reviewer rightly identifies, this manuscript does not has an independent "discussion" section. We chose to go with the alternate "results and discussion" structure, which is indicated as acceptable in the instructions for authors for this journal. The rationale is that our work, as we have mentioned before, is mostly of a descriptive nature, in which the behavior of the early stage of the fourth wave (dominated by the omicron variant) in Mexico City is shown. As such, we consider that the structure of showing the results and briefly discussing their implications in the same section is more straightforward for the type of analyses performed.

Point 7. There are some typo or labeling issues, eg, Figure4 figure legend. “In spike proteinspikes” should be “in spike protein”. There I s no labeling for Figure 8 panel A-C.

Response: The typo has been corrected (line 386) and the caption of figure 8 has been modified (line 536-528).

Reviewer 2 Report

1. Authors have mentioned that new variant omicrons have features that can adapt and escape from the immune system. If the author could discuss what kinds of adaptation have been associated with omicron, it could be more informative. 2. Provide the information of the total number of samples collected for this study. 3. Is the short duration of sample collection enough to analyze the epidemiological profile of the omicron variant? 4. In addition to the report if the author provides the data of the selected patient's history, co-infections in table form could be easy for the readers and also be more informative. 5. Authors have mentioned that in the 50th week, the omicron variant increased rapidly and reached 65%; where all the 65% of infection is caused by a single variant or it is a double variant? 6. Did other mutations in the region of other parts of the spike protein also increase the transmission or only the mutations occur if RBD is responsible for the same? 7. Increase the figure quality. 8. Also, can the author discuss the complete mutational details of all variants in a table. So future researchers can refer

Author Response

Response to Reviewer 2 comments

Point 1. Authors have mentioned that new variant omicrons have features that can adapt and escape from the immune system. If the author could discuss what kinds of adaptation have been associated with omicron, it could be more informative.

Response: Thank you for your comment. With the word “adaptation” we wanted to refer to the high transmissibility and immune escape of the variants of concern, to make it clearer we decided to remove it. In addition we added a very current reference describing the immune escape of the variants of concern. This has been modified in line 57.

Point 2. Provide the information of the total number of samples collected for this study.

Response: The number of samples used in each analysis is added in the methodology section, lines 133, 244 and 270.

Point 3. Is the short duration of sample collection enough to analyze the epidemiological profile of the omicron variant?

Response: The goal of this manuscript is to provide an early description of the fourth wave in Mexico City, currently ongoing, which is dominated by the omicron variant. In this regard, the analyzed time frame comprises the period from the introduction of the variant, to it becoming the dominant (as detected by genomic surveillance). With this in mind, we agree that a full description of Omicron's epidemiological profile is not possible with just this data; however, we should emphasize that this was not the intent of the work. Rather, we were interested in having the most complete description of this early time frame, in which both Delta and Omicron variants coexisted in Mexico City, to search for signals of changes in (population or individual) risk associated with Omicron.

Point 4. In addition to the report if the author provides the data of the selected patient's history, co-infections in table form could be easy for the readers and also be more informative.

Response: We appreciate this comment. We have incorporated a supplementary file containing the anonymized data of these cases discussed in the manuscript. (lines 493-494).

Point 5. Authors have mentioned that in the 50th week, the omicron variant increased rapidly and reached 65%; where all the 65% of infection is caused by a single variant or it is a double variant?

Response: The percentage of BA.1 and BA.1.1.1 lineages detected in weeks 46th, 50th and 52th, has been detailed in lines 320-323.

Point 6. Did other mutations in the region of other parts of the spike protein also increase the transmission or only the mutations occur if RBD is responsible for the same?

Response: Thank you for the important observation, so far no mutation found outside the RBD has been associated with increased virus transmission. We focus on mutations within the RBD because of their importance in binding to the ACE2 receptor and their impact on transmissibility.

Point 7. Increase the figure quality. 8. Also, can the author discuss the complete mutational details of all variants in a table. So future researchers can refer.

Response: Figure 3 quality has been increased. A supplementary table with the details of the mutations of 783 omicron samples analyzed in this work was carried out (lines 348-356).

Reviewer 3 Report

This article intends to investigate the genomic surveillance of the SARS-CoV2 Omicron variant in Mexico City. They have found that the R346K substitution is associated with immune escape by monoclonal antibodies. In the phylogenetic analysis, they found several independent exchanges between Mexico and the world. The haplotype analysis revealed that there was no association between haplotype and vaccination status. 66% of patients were vaccinated, none of the reported comorbidities were associated with Omicron, the presence of odynophagia and absence of dysgeusia were significant predictor symptoms for Omicron, and the Ct value on RT–qPCR was lower in Omicron. This manuscript timely provided the information regarding the Omicron variant.

Several suggestions:

  1. Line 94, what is [n patients]?
  2. Line 114, please add a reference (e.g., website) after [following the manufacturer’s instructions].
  3. Line 145, what is [VOC ~ Feature_1+ Featu〖re〗_(2 )+ ...+Feature_i]?
  4. Line 159, what is [nucmer]?
  5. Lines 387-388, [observation of a dropout of the S-gene target associated with the Omicron SARS-CoV-2 variant.] means less S-gene is detected. However, Ct of detection of Spike gene is low in the Omicron variant in Figure 8 (right panel), why?

Author Response

Response to Reviewer 3 comments

Point 1. Line 94, what is [n patients]?

Response: The character n in line 96 was eliminated.

Point 2. Line 114, please add a reference (e.g., website) after [following the manufacturer’s instructions].

Response: The reference was added in line 119.

Point 3. Line 145, what is [VOC ~ Feature_1+ Featu〖re〗_(2 )+ ...+Feature_i]?

Response: We use the formula tool in word to write it clearly. The equation represents the logistic model in using the "formula" notation (common in statistical software) to represent that the Variant of Concern label (delta or omicron) is a function of the set of features in the model (either comorbidities or symptoms). As we mentioned before, the actual model being used is a logistic regression. Lines 264-265.

Point 4. Line 159, what is [nucmer]?

Response: NUCmer is a Perl script pipeline for the alignment of multiple closely related nucleotide sequences, this line was added in line 140.

Point 5. Lines 387-388, [observation of a dropout of the S-gene target associated with the Omicron SARS-CoV-2 variant.] means less S-gene is detected. However, Ct of detection of Spike gene is low in the Omicron variant in Figure 8 (right panel), why?

Response: Thank you for your comment, we have updated the right panel of Figure 8, since in the previous one we were plotting the value of 0 to identify the samples that did not present amplification in the Spike gene, which made it show a non-real increase of samples in these values.

The new graph does not take into account the zero values. We note that of the omicron samples, only eight had amplification in the spike gene with Ct values 16 to 33, of which five samples have a true exponential amplification curve in the spike gene in the PCR, even though they have the 69-70 deletion in spike detected by sequencing. The remaining 3 samples did not have a true amplification curve for the spike gene in PCR. This information was added in lines 507-512.

Round 2

Reviewer 1 Report

The authors have responded effectively to previous comments and concerns by text edits and explanations. This is much appreciated. I have no further comments or concerns.

Reviewer 2 Report

The revised version can be accepted and recommend to get published. 

Best wishes